Social-aware trajectory prediction using goal-directed attention networks with egocentric vision

http://orcid.org/0000-0001-8424-5758 Astuti Lia
Chiu Chui-Hong
http://orcid.org/0000-0002-6735-0473 Lin Yu-Chen yuchlin@fcu.edu.tw
http://orcid.org/0000-0001-7701-6965 Lin Ming-Chih
Automatic Control Engineering, Feng Chia University , Taichung , Taiwan
Benítez-Andrades José Alberto
Electronic publication date: 2025 Apr 25
Publication date: 2025
Volume: 11
Electronic Location ID: e2842
Received 2024 Oct 25; Accepted 2025 Mar 27
Copyright: © 2025 Astuti et al.
Copyright year: 2025
Copyright holder: Astuti et al.
License: This is an open access article distributed under the terms of the Creative Commons Attribution License, which permits unrestricted use, distribution, reproduction and adaptation in any medium and for any purpose provided that it is properly attributed. For attribution, the original author(s), title, publication source (PeerJ Computer Science) and either DOI or URL of the article must be cited.
License URL: https://creativecommons.org/licenses/by/4.0/

Keywords: Social-goal attention networks, Heterogeneous road users, Multimodal trajectory prediction, Socially-aware interaction correlation, Goal-directed forecaster

Funding: National Science and Technology Council, Taiwan 113-2218-E-035-001 & 112-2628-E-035-001-MY3 This work was supported by the National Science and Technology Council, Taiwan, R.O.C. (No. 113-2218-E-035-001 & 112-2628-E-035-001-MY3). The funders had no role in study design, data collection and analysis, decision to publish, or preparation of the manuscript.

==============================
This study presents a novel social-goal attention networks (SGANet) model that employs a vision-based multi-stacked neural network framework to predict multiple future trajectories for both homogeneous and heterogeneous road users. Unlike existing methods that focus solely on one dataset type and treat social interactions, temporal dynamics, destination point, and uncertainty behaviors independently, SGANet integrates these components into a unified multimodal prediction framework. A graph attention network (GAT) captures socially-aware interaction correlation, a long short-term memory (LSTM) network encodes temporal dependencies, a goal-directed forecaster (GDF) estimates coarse future goals, and a conditional variational autoencoder (CVAE) generates multiple plausible trajectories, with multi-head attention (MHA) and feed-forward networks (FFN) refining the final multimodal trajectory prediction. Evaluations on homogeneous datasets (JAAD and PIE) and the heterogeneous TITAN dataset demonstrate that SGANet consistently outperforms previous benchmarks across varying prediction horizons. Extensive experiments highlight the critical role of socially-aware interaction weighting in capturing road users’ influence on ego-vehicle maneuvers while validating the effectiveness of each network component, thereby demonstrating the advantages of multi-stacked neural network integration for trajectory prediction. The dataset is available at https://usa.honda-ri.com/titan.

Introduction

Safe navigation in complex traffic scenarios is fundamentally dependent on accurately predicting the future trajectories of diverse road users. In autonomous and semi-autonomous driving systems, reliable trajectory prediction is crucial for mitigating safety risks arising from the unpredictable behavior of pedestrians, cyclists, vehicles, and other road users (Golchoubian et al., 2023; Fang et al., 2024; Sun et al., 2025). Despite significant progress in the literature, existing approaches exhibit three primary limitations that hinder their practical application in trajectory prediction under egocentric vision scenarios.

First, much of the prior work has focused on trajectory prediction from a bird’s-eye view, typically acquired from high-altitude cameras (Geng et al., 2023; Lan et al., 2024). Models such as Social-long short-term memory (LSTM) (Alahi et al., 2016), Social-generative adversarial networks (GAN) (Gupta et al., 2018), and subsequent graph-based approaches (Schöller et al., 2020; Wang et al., 2023) are predominantly pedestrian-centric and designed for static spatial layouts. Consequently, these methods do not address the unique challenges associated with egocentric vision, where sensor data from dashboard cameras and inertial measurement units (IMUs) or on-board diagnostic (OBD) introduce additional complexities, including dynamic observer-centric biases and the need for synchronized multi-modal inputs. Second, although recent multimodal trajectory prediction frameworks, such as Bi-directional Pedestrian Trajectory Prediction (BiTraP) (Yao et al., 2021) and ActionBased Contrastiv (ABC+) (Halawa, Hellwich & Bideau, 2022), have advanced uncertainty estimation through probabilistic forecasting. Dong et al. (2024) introduced Two-stream Sparse-character-based Network (TSNet) to filter negative pedestrian character information in first-person view, while Hu et al. (2025) developed Probabilistic Multimodal Trajectory Prediction Network (PMTPN) for probabilistic Vulnerable Road User (VRU) trajectory forecasting using multimodal inputs. Feng et al. (2024) employed a transformer-based model for inconspicuous pedestrian movements, and Astuti et al. (2024) proposed CITraNet, leveraging a Transformer-based Gumbel distribution network to predict VRU trajectory and crossing intentions. Although these methods enhance pedestrian and cyclist predictions, they do not extend to heterogeneous traffic environments involving cars, motorcyclists, and other road users. A further gap in the literature is the absence of unified architectures that integrate heterogeneous data sources and capture interactions among diverse road users. Existing methods often rely on single-module solutions that separately encode spatial, temporal, or goal-directed forecaster, uncertainty features. Zhao & Wildes (2021) introduced a goal-conditioned expert and Wang et al. (2022a) proposed a stepwise goal-driven network to enhance goal estimation, but they did not integrate spatial and temporal dependencies simultaneously. Zhou et al. (2023b) developed Conditional variational autoencoder module and a Socially-aware Regression (CSR), a cascaded Conditional Variational Autoencoder (CVAE) with socially-aware regression, which reduces error accumulation but does not fully leverage multi-sensor fusion, and Zhang & Li (2022) proposed Multimodal Trajectory Prediction Transformer (MTPT) to enhance explainability through attention mechanisms but did not jointly model social interactions and uncertainty. These limitations underscore the need for a comprehensive framework that integrates multimodal fusion, social interactions, goal-directed forecaster, and uncertainty modeling in egocentric vision scenarios.

To address these gaps, we propose a novel social-goal attention networks (SGANet) model that leverages vision-based multi-stacked neural network integrations. This approach unifies spatial, temporal, and goal-directed modeling, along with uncertainty estimation, in a single coherent framework. SGANet processes trajectory from dashboard camera and odometry data from inertial measurement unit (IMU) or on-board diagnostic (OBD) sensors through four key modules: Encoder module: A graph attention network (GAT) adaptively prioritizes interactions based on road user class, spatial proximity, and trajectory intent, allowing differential weighting of agents in the scene. For instance, pedestrians at crosswalks exert a higher influence on trajectory adjustments compared to distant vehicles, while cyclists and motorcyclists require adaptive weighting due to their dynamic maneuverability. A long short-term memory (LSTM) network encodes complex temporal dependencies to capture sequential trajectory patterns across different road users.

Goal-Directed Forecaster (GDF) Module: GDF predicts coarse future goals by estimating goal distributions, providing an intermediate representation that guides subsequent trajectory refinement. Including a goal-directed mechanism addresses the issue of partial forecasting, where long-term intentions and intermediate targets of different agent types are often overlooked.

Conditional Variational Autoencoder Module: By modeling the inherent uncertainty in road user behavior, the CVAE module generates multiple plausible trajectory predictions as socially-aware correlation interaction model. This module is critical for accounting for the varied behavior of road users and addresses the limitation of unimodal or deterministic trajectory predictions.

Decoder Module: Integrated multi-head attention (MHA) and feed-forward networks (FFN) refine the decoded trajectories, ensuring that both short-term and long-term dynamics are accurately captured. This module resolves the need for an end-to-end trajectory model integration of all features, including social interactions and uncertain goal predictions.

By integrating social interactions, temporal dependencies, goal-directed forecasting, and multimodal uncertainties, SGANet enhances trajectory prediction for both homogeneous and heterogeneous road users in egocentric vision scenarios. Evaluations on the pedestrian-focused datasets JAAD (Rasouli, Kotseruba & Tsotsos, 2017) and PIE (Rasouli et al., 2019) demonstrate SGANet’s generalization to vulnerable road users. In contrast, the TITAN dataset (Malla, Dariush & Choi, 2020), which includes diverse road users such as pedestrians, cyclists, and vehicles, is employed to assess the model’s performance in heterogeneous traffic scenarios. Results indicate that SGANet outperforms previous benchmarks across prediction horizons of up to seven seconds, confirming its effectiveness in varied traffic environments. In summary, the main contributions of this study are as follows: 1. This study introduces a novel social-goal attention network (SGANet) as vision-based multi-stacked neural network that unifies spatial, temporal, goal-directed modeling, and uncertainty estimation for trajectory predictions.

2. SGANet considers socially-aware interaction correlation, goal-directed forecaster, and multimodal modelling, which inputs sequential data from two key different sensors system, such as dashboard-mounted car camera that captured road user sequential data and IMU/OBD sensor that captured accelerometers and gyroscopes of the ego-vehicle.

3. Extensive evaluations on homogeneous (JAAD and PIE) and heterogeneous TITAN datasets demonstrate superior predictive trajectory accuracy across varying horizons, validating SGANet’s effectiveness in egocentric vision and mixed traffic environments.

Related works

Recent advances in trajectory prediction can be categorized into two key areas: (1) Bird’s-eye view and pedestrian-focused trajectory prediction models, (2) Vision-based multi-stacked neural network models integrating spatial, temporal, goal-directed forecaster, and uncertainty modeling.

Bird’s-eye view and pedestrian-focused trajectory prediction models

Early trajectory prediction works primarily employed bird’s-eye view imagery to model pedestrian behavior. Alahi et al. (2016) introduced Social-LSTM, which uses recurrent neural networks (RNN) with a social pooling layer to capture interpersonal interactions. Dendorfer, Osep & Leal-Taixé (2020) proposed Goal-GAN, which decomposes prediction into goal estimation followed by trajectory refinement. Zhang & Li (2022) introduced the multimodal trajectory prediction transformer (MTPT) and Huang et al. (2019) introduced STGAT to fuse spatial and temporal interactions via GAT mechanisms. The availability of egocentric pedestrian datasets such as JAAD (Rasouli, Kotseruba & Tsotsos, 2017) and PIE (Rasouli et al., 2019) further enabled researchers to analyze pedestrian dynamics from a moving vehicle’s perspective. Subsequent frameworks like BiPed (Rasouli, Rohani & Luo, 2021) extended these approaches by jointly predicting pedestrian trajectories and actions, while methods such as the multi-information CNN by Wang et al. (2021) integrated historical trajectories and contextual cues to improve prediction accuracy.

To address inherent uncertainties in trajectory prediction, several multimodal frameworks have been developed (Li et al., 2024b; Ruan et al., 2024). Approaches such as ABC+ (Halawa, Hellwich & Bideau, 2022), SGNet (Wang et al., 2022a), and BiTraP (Yao et al., 2021) leverage CVAE to generate diverse trajectory hypotheses and quantify uncertainty. Complementary methods include the non-parametric deep ensemble approach by Li et al. (2024a). Huynh & Alaghband (2021) proposed GPRAR to reconstruct human poses and actions under noisy conditions, while Liu et al. (2024) introduced a two-stage method for modeling interpretable social interactions, and Lv et al. (2023) developed SSAGCN using social soft attention to simultaneously capture pedestrian interactions and environmental context. These studies collectively advance multimodal prediction but remain largely confined to homogeneous, pedestrian-centric scenarios.

Recent works have focused on refining trajectory prediction from an egocentric perspective by employing advanced feature extraction and goal estimation techniques. Dong et al. (2024) introduced TSNet to mitigate negative pedestrian character information, and Hu et al. (2025) developed PMTPN for probabilistic forecasting of vulnerable road user trajectories using multimodal inputs. Feng et al. (2024) employed a transformer-based forward generation model, while Astuti et al. (2024) presented CITraNet, which uses a Transformer and extreme value theory integration to predict trajectory and crossing intentions. Complementary to these, Chen et al. (2023) proposed a stacking model for real-time trajectory prediction of vulnerable traffic participants, Lin, Lin & Hung (2023) focused on predicting potentially dangerous pedestrian behavior with an Attention-LSTM architecture, Rasouli & Kotseruba (2023) introduced PedFormer for cross-modal pedestrian behavior prediction, and Sharma, Dhiman & Indu (2023) developed a visual–motion–interaction-guided framework for pedestrian intention prediction. Although these approaches improve egocentric predictions for pedestrians and bicyclists, they do not fully address heterogeneous traffic environments that include vehicles and motorcyclists, thereby motivating the need for more comprehensive models.

Vision-based multi stacked neural networks trajectory prediction models

Different from most studies that rely on a single encoder–decoder architecture for trajectory prediction, the present study leverages a vision-based multi-stacked neural network framework that integrates several specific modules to learn from sequential sensor data. The literature in this area can be broadly categorized into the following five components: 1. Graph-based models: Socially aware encoder architectures employing graph convolutional networks (GCN) (Huynh & Alaghband, 2021; Lv et al., 2023; Wang et al., 2022b) and graph neural networks (GNN) (Li et al., 2022a) have been widely used in edge graph trajectory prediction. Recently, GAT (Veličković et al., 2018) has emerged as a superior alternative, as it can adaptively learn more comprehensive graph representations than conventional GCN or GNN. For example, GraphTCN (Wang, Cai & Tan, 2021) utilizes a CNN-based spatial–temporal graph framework to capture edge features with salient spatial information, and STGAT (Huang et al., 2019) assigns importance to neighboring nodes to generate spatial interactions. In light of these developments, our framework incorporates a GAT network to implicitly assign spatial interaction weights within the encoder module.

2. LSTM-based models: Classic LSTM networks address the vanishing gradient problem encountered in RNNs. Social-LSTM (Alahi et al., 2016) aggregates hidden states of pedestrians using social pooling, and Gupta et al. (2018) combine LSTM cells with generative adversarial (GAN) networks. Moreover, LSTM networks have proven effective in encoding temporal information by preserving long-term dependencies in complex sequential sensor data (Chen et al., 2023; Sharma, Dhiman & Indu, 2023; Li et al., 2020). Our approach leverages LSTM networks to robustly encode the temporal dynamics present in the data.

3. Goal-based models: Recent research has incorporated goal networks to condition trajectory predictions. For instance, Goal-GAN (Dendorfer, Osep & Leal-Taixé, 2020) investigates multimodal conditioning routes for estimating final positions, ExpertTraj (Zhao & Wildes, 2021) generates predictions conditioned on goals derived from an expert repository, and GTP (Tran, Le & Tran, 2021) employs a dual-channel neural network to integrate goal information into future trajectory estimation. Additionally, BiTraP (Yao et al., 2021) and SGNET (Wang et al., 2022a) estimate multimodal goals. Motivated by these approaches, our framework includes a GDF module to generate successive goal estimation.

4. CVAE-based models: CVAE-based models have been extensively used for multimodal trajectory prediction, particularly in capturing individual driving styles and uncertainty in human behavior. ABC+ (Halawa, Hellwich & Bideau, 2022) employs an action-based contrastive learning loss, while AgentFormer (Yuan et al., 2021) infers agents’ latent intent to model social influence. CSR (Zhou et al., 2023b) enhances pedestrian prediction through cascaded sequential estimation, whereas DACG (Zhou et al., 2023a) refines human intention modeling by incorporating time-dependent social interactions. Importantly, CVAEs can condition trajectory generation on driver-specific attributes such as reaction times and maneuver tendencies, making them well-suited for personalized trajectory forecasting (Liu et al., 2024). Unlike graph-based networks, which primarily encode spatial and temporal dependencies, CVAEs enable diverse trajectory hypotheses that better capture individual motion variations under uncertainty (Chen et al., 2023). Recognizing these advantages, our study integrates a CVAE module to generate multimodal trajectories across homogenous and heterogeneous road users while considering driver-specific characteristics.

5. Attention-Based Models: Recent work has demonstrated that attention mechanisms enhance trajectory prediction by effectively mixing and regulating information. Lv et al. (2023) employed soft attention with GCN, and Li, Wang & Zuo (2023) integrated MHA and FFN into encoder and social interaction modules. Li et al. (2024b) further combined historical motion data with environmental context. These developments motivate our inclusion of MHA and feed-forward networks in both the decoder and the goal-directed forecaster (GDF) modules.

This study extends these advancements by integrating graph-based, LSTM-based, goal-directed forecaster, CVAE-based, and attention-based modules within a unified vision-based multi-stacked neural network framework. This comprehensive framework explicitly models socially aware interaction correlations and multimodal trajectory uncertainties for both homogeneous and heterogeneous road users.

Problem formulation

This study aims to predict future trajectories for multiple road user classes by integrating multimodal trajectory prediction and socially-aware interaction correlation. We validate our approach on both heterogeneous and homogeneous datasets. The TITAN dataset (Malla, Dariush & Choi, 2020) captures diverse road users using a GoPro Hero 7 and an IMU sensor. The PIE (Rasouli et al., 2019) and JAAD (Rasouli, Kotseruba & Tsotsos, 2017) datasets focus on pedestrian behavior using mounted dashboard cameras and OBD sensors. The following sections outline the input processing for each dataset.

Multimodal input information sequences

The proposed model processes multimodal input sequences using a feature extractor, as illustrated in Fig. 1. The detection and tracking system assign labels to all road users after recording traffic scenarios using the dashboard camera. Road user categories: Five road user types are considered, such as pedestrians, bikers, cyclists, trucks, and cars.

Bounding box representation: Each road user’s trajectory is defined by its bounding box coordinates, which include top-left {xtl,ytl} and bottom-right {xbr,ybr} corners.

Speed estimation: The speed of each road user is incorporated to model interaction weights and influence levels.

Distance measurement: The distance between the ego-vehicle and each road user.

Figure 1 The overall architecture of the proposed social-goal attention networks (SGANet) model.

The model consists of four main modules: encoder, goal-directed forecaster (GDF), conditional variational autoencoder (CVAE), and decoder.

Since this study focuses on egocentric vision, odometry data from IMU and OBD sensors captures the ego-vehicle’s motion state over time. The accelerometer records longitudinal acceleration along the x-axis for forward movement (m/s2), while the gyroscope measures yaw velocity (rad/s) alongInt the y-axis to track turning behavior. To ensure temporal synchronization, multimodal input streams are aligned using timestamps specific to each dataset. TITAN synchronizes camera and IMU sensor readings at 10 Hz, while PIE and JAAD synchronize camera and OBD sensor data at 30 FPS. Since different modalities operate at different frequencies, sequential frame alignment is performed to match sensor readings with their corresponding visual frames, ensuring consistency across input streams. These multimodal inputs are then concatenated into a unified sequence:

(1) Xt={bt,st,dt,ot}

where bt represents road users’ classes ct and bounding box coordinate. st is the speed of each road user as st={xtl,brtc−xtl,brtc−1,ytl,brtc−ytl,brtc−1} at current frame tc. dt denotes the distance between ego-vehicle o and road user n in a scene, which are calculated into dt={(xn−xo),(yn−yo)}. ot is odometry information as the state of ego-vehicle over the time.

Historical and future time sequences

At current frame t, the historical information of each target road user n is given as:

(2) Hnt={Xtobs,...,Xt}

where Hnt is the observed historical sequence, Xt represents the concatenated input, and t is the maximum historical sequence length. The objective is to predict future trajectories as:

(3) Ynt={bt+1,bt+2,...,bt+Tf}

where Ynt represents the future bounding box trajectory for each of road user n over a prediction horizon Tf frames.

Feature extractor

Each input sequence is embedded into a 512-dimensional feature vector using a fully connected (FC) network, ensuring a structured representation for downstream processing:

(4) entobs=ψemb(X;Wemb)

where ψemb is the embedding sub-layer with learned weight matrix Wemb. Following this, the proposed SGANet processes these embeddings to learn multimodal interactions and predict future trajectories up to the maximum prediction horizon Tf.

Methods

Spatial interaction encoder module

Recent studies emphasize the importance of modeling interactions between road users and their neighbors (Huynh & Alaghband, 2021; Li et al., 2022a; Veličković et al., 2018; Wang et al., 2021). Our method models each road user as a node in a fully connected graph, where edges represent interaction dependencies. Figure 2, based on the TITAN dataset (Malla, Dariush & Choi, 2020), illustrates the complete GAT network for social interactions, while Fig. 3 provides a focused view of a single GAT layer, specifically highlighting interactions centered on node h1. This distinction clarifies the broader interaction framework versus node-specific attention modeling, offering a comprehensive perspective on the spatial interaction encoder.

Figure 2 The GAT’s full network of social interactions.

At each time sequence, multiple road user classes in a scene are represented as nodes in a complete graph, with edges depicting their interactions.

Figure 3 A zoomed-in view of a single GAT layer.

(A) illustrates a graph attention layer in a real traffic scenario, while (B) side shows an actual GAT representation where a node assigns different levels of importance to its closest neighbors and aggregates features obtained from these nodes.

Our model employs GAT to quantify social interactions by dynamically assigning influence weights to different road user classes. The attention mechanism prioritizes interactions based on proximity, speed, trajectory direction, and class-specific behavioral tendencies, allowing the model to weigh certain road users more heavily in trajectory prediction. For instance, faster-moving vehicles receive higher attention scores when approaching a pedestrian’s trajectory, while stationary objects receive minimal weighting. Additionally, interactions between pedestrians and nearby vehicles are weighted more heavily than interactions between two distant vehicles, ensuring that socially relevant behaviors are emphasized.

Each node in the GAT layer is represented by the input feature h={h⇀1,h⇀2,...,h⇀N}, where h⇀N∈RF and F is the feature dimension of each node. Given an observed frame, the GAT layer receives input with htobs={btobs,stobs}. The coefficients between node i and its neighboring node j can be computed by:

(5) αijtobs=exp(ELU(aT[Whitobs∥Whjtobs]))∑k∈N(i)exp(ELU(aT[Whitobs∥Whktobs]))

where αijtobs is the attention coefficient of node j to i at the observed time sequences tobs, Ni represents the neighbors of node i on the graph, the normalization is using softmax with ELU activation function, a∈RF is weight vector of single FFN layer, W is the learned weight matrix of inputted information sequence embedding htobs∈entobs and a shared linear transformation which is applied to each node, and ∥ is the concatenation operation.

Once the attention coefficients are computed, the updated representation of node i in the GAT layer is obtained as:

(6) Aitobs=σ(∑j∈N(i)αijtobsWhjtobs)

where σ represents nonlinear activation functions. Equations (5) and (6) describe a single GAT layer to aggregate socially aware interaction features. Aitobs is the result of the aggregated hidden state and contains the socially-aware correlation interaction from the target road user i.

Temporal encoder module

As shown in Fig. 4, the proposed encoder module utilizes an LSTM network to capture temporal dependencies, where solid and dashed arrows indicate the current time tc and the observation time sequences tobs, respectively. After extracting the object feature entobs and spatial interaction encoding Antobs for each road user n, these representations are concatenated along with the aggregated goal information Ygoaltobs. The goal feature encoder, which encodes the bounding box ground truth at the maximum prediction horizon Tf, is derived from the GDF module and serves as the goal feature at the current time step.

Figure 4 Temporal encoder.

The overall structure of the encoding process captures spatial interactions, goal features, and object features from consecutive frames. Solid arrows represent the current time step, while dashed arrows indicate the observation time sequence.

Given a road user n in a scene, the object feature embedding entobs, spatial interaction encoding from the GAT layer Antobs, and goal feature Ygoaltobs are concatenated and fed into an LSTM cell. This process is formulated as:

(7) menctobs=LSTM(concat(Antobs,entobs,Ygoaltobs),mntc−1;Wm)

where menctobs represents the hidden state of the LSTM cell at the current time step tc up to the observation state tobs with its learned weight matrix Wm. The initial hidden state Ygoaltobs is set to a zero vector. These parameters are shared across all road users in the scene, and the encoder output menctobs is subsequently fed into the GDF and CVAE modules.

Goal-directed forecaster module

The GDF module generates coarse-to-fine goals for trajectory prediction and refines future goals at each time step. Figure 5 illustrates its overall structure, where solid and dashed arrows indicate current tc and predicted tpre time sequences. Given a predicted goal bounding box location (xtlTf,ytlTf,xbrTf,ybrTf) at the maximum future time Tf, with the initial goal set as a zero-filled vector t=0. The GDF module predicts coarse goals, which are passed to the encoder and CVAE modules, generating goal loss. This study uses twenty noisy goals to menct at each time step t to evaluate their impact on prediction accuracy before being processed by the CVAE module.

Figure 5 The overall structure of the goal-directed forecaster (GDF) module.

Solid arrows represent the current time step, while dashed arrows indicate the prediction time sequence.

A set of observed goal locations Ygoaltobs is concatenated with the input to enhance the temporal encoder menctobs, enabling it to learn discriminative representations across timsequences. At each future step tc+tpre, the CVAE module utilizes a subset of consecutive goals Ygoaltc+tpre as coarse guidance for trajectory prediction. The goal regressor YregTf compresses multiple goals into a single representation, optimizing the goal loss function.

To improve prediction accuracy, a goal aggregator using soft attention is introduced to weigh successive goals before passing them to the encoder and CVAE modules. The input data menctobs is reshaped to match the goal embedding size using a linear transformation followed by ELU activation:

(8) mgtc+1=ELU(Wgmenctobs+bg)

where mgtc+1 represents hidden goals at time tc+1 and b is the bias term. The sequence mgtc+1 is then processed by an LSTM cell, defined as:

(9) mgtc+tpre=σg(LSTM(mgtc+1,mgtc+tpre;Wg)).

The MHA network is applied to refine consecutive hidden goal inputs mgtc+tpre, where mgtc+tpre={mgtc+1,...,mgtc+tpre,...,mgtc+Tf}. Given queries Q, keys K, and values V from the hidden goal network mgtc+tpre, each attention head computes:

(10) Att(Qh,Kh,Vh)=softmax(QKT/dk)V

where dk is the scaling factor, and each head has its own weight matrices such as Qh=QWQh, Kh=KWKh, Vh=VWVh. Then, the MHA network’s output Att(Qh,Kh,Vh) is passed through a soft attention mechanism α, which assigns attention weights to lth predicted goal scores that can be written as:

(11) αl=exp(eT)∑n=1Nexp(eT)

where eT=aT(WaAtt(Qh,Kh,Vh)+ba). The computed attention weights αl are then summed with the normalized weights, producing observed goal location Ygoaltobs and predicted goal location Ygoaltc+tpre. These are passed to the encoder and CVAE modules, as follows:

(12) Ygoaltobs=softmax(ELU(Wencαl+benc))

(13) Ygoaltc+tpre=softmax(ELU(Wdecαl+bdec)).

Finally, to regularize the GDF module, we regress YregTf to infer a single coarse goal representation, defined as:

(14) YregTf=ϕreg(Att(Qh,Kh,Vh);Wreg).

Conditional variational autoencoder module

This study adopts CVAE for multimodal trajectory prediction. Figure 6 illustrates the CVAE module structure, where the red dashed line indicates operations using ground truth during training, the solid blue line represents common flows in both training and testing, and the solid green line is exclusive to testing. The CVAE module consists of three sub-networks: 1. The conditional prior network pθ(Z|menctobs)

2. The recognition network qϕ(Z|menctobs,E)

3. The trajectory generation network pω(mLatenttc+tpre|menctobs,Ygoaltc+tpre,Z)

Figure 6 The overall structure of the conditional variational autoencoder (CVAE) module.

The red dashed line represents the operation flow using ground truth during training. The solid blue line indicates the operation flow in both training and testing, while the solid green line represents the operation flow during testing.

where θ, ϕ, and ω represent their respective network parameters, Ygoaltc+tpre is GDF module output (from Eq. (9)) and menctc−tobs denotes the encoder module output (from Eq. (4)). All networks are implemented using FC layers.

CVAE module during training process

The recognition network qϕ(Z|menctobs,E) captures dependencies between the observed trajectory state menctobs and the ground truth trajectory E. The ground truth trajectory E is processed by a bidirectional LSTM (Bi-LSTM) to generate the hidden state mEtc+tpre, which computed as:

(15) mEtc+tpre=Bi−LSTM(E,mEtc+1;WE)

where E is the target ground truth trajectory {entc+1;...;entc+tpre;...;entf} at future time sequence tc+tpre for each road user n, and en consists of coordinate position bt, distance dt, speed st, and odometry ot. The recognition network does not assume a predefined distribution for E but implicitly learns it through the distribution Zq∈Z and Zq calculation is as follows:

(16) Zq=qϕ(concat(menctobs,mEtc+tpre);Wq)

where μZq and σZq represent the predicted distribution mean and covariance to captures the dependencies between the observed trajectory menctobs and the ground truth mEtc+tpre.

CVAE module during testing process

During inference, the prior network pθ(Z|menctobs) samples multiple latent variables Z to generate diverse trajectory predictions. This enables the model to capture different driving styles, reaction times, and trajectory patterns, ensuring the predicted trajectories reflect individual variations in behavior. By leveraging multimodal predictions, CVAE accounts for uncertainties in driver decision-making, such as varying acceleration tendencies or reaction delays to obstacles.

Moreover, the prior network pθ(Z|menctobs) operates without knowledge of the ground truth E. Instead, it predicts the distribution mean μZp and covariance σZp using a subset of Z, which consists of the predicted goal location Ygoaltc+tpre and the encoder input menctobs. Multiple trajectory samples are drawn from N(μZp,σZp) to generate the latent variable Zp, formulated as:

(17) Zp=pθ(concat(menctobs,Ygoaltc+tpre);Wp).

To generate multimodal trajectory predictions Yntc+tpre, the trajectory generation network mLatenttc+tpre utilizes the latent space by concatenating samples from N(μZp,σZp) and N(μZq,σZq) along with menctobs and Ygoaltc+tpre, expressed as:

(18) mLatenttc+tpre=pω(concat(menctobs,Ygoaltc+tpre,Z);Wp).

CVAE module uses Kullback-Leibler divergence loss

To address the absence of the ground truth target E during testing, the Kullback-Leibler (KL) divergence loss (DKL) between N(μZp,σZp) and N(μZq,σZq) is minimized. This ensures that the prior network pθ(Zp|menctobs) implicitly learns the dependency between the observed trajectory menctobs and the future trajectory E. The KL loss optimization is formulated as:

(19) DKL[qϕ(Zq|menctobs,E)∥pϕ(Zp|menctobs)]=∑Z∈Kqϕ(Zq|menctobs,E)logpθ(Zp|menctobs)qϕ(Zq|menctobs,E).

Decoder module

The decoder module predicts the trajectory distribution from the CVAE latent space mLatenttc+tpre. Figure 7 illustrates the overall decoder structure, which consists of a MHA, FFN, and FC networks. The solid and dashed arrows represent the current time step tc and prediction time sequences tc+tpre, respectively. To decode the predicted trajectory Y, where Y=[Yntc+1;...;Yntc+tpre;...Yntc+Tf] represents the predicted future bounding box coordinate location for each road user. The decoder module leverages MHA to capture diverse relationships within the input data. Based on Eq. (10), for each attention head h, the latent space mLatenttc+tpre from the CVAE module is linearly transformed into Qh,Kh,Vh and the attention score is computed using the scaled dot product of queries Qh and keys Kh.

Figure 7 The overall structure of the decoder module with the autoregressive strategy.

The solid arrows represent the current time sequence, while the dashed arrows indicate the prediction time sequence.

The FFN then processes the output of MHA, transforming it into a new hidden state mffntc+1 for the next time step tc+1 using layer normalization, formulated as:

(20) mffntc+tpre=LayerNorm(mffntc+1+Att(Qh,Kh,Vh)).

Finally, the trajectory regressor, a single FC layer, takes mffntc+tpre as input to compute the trajectory prediction Y of each road user n at the next future time step tc+tpre, expressed as:

(21) Yntc+tpre=ψdec(Yntc+1;Wdec).

Loss functions

This study adopts the best-of-many (BoM) approaches to minimize the distance between the best predicted trajectory and the ground truth trajectory, where Y^=[Y^ntc+1;...;Y^ntc+tpre;...;Y^ntc+Tf] represents the ground truth at future time sequences tc+tpre and Tf is the maximum future time. Multimodal trajectory prediction enhances model accuracy and diversity by capturing data variation. Root mean squared error (RMSE) is used as the loss function for the encoder module LTraj.

To ensure accurate goal network trajectory predictions, the goal prediction loss LGoal is optimized using RMSE between the predicted goal YregTf and the ground truth goal Y^Tf. The KL loss (DKL) is applied to the CVAE module LCVAE to reduce differences between the recognition and prior networks. The total loss function for each of training iteration is given by:

(22) Ltotal=LTraj+LGoal+LCVAE.

Each loss function is formulated as follows:

(23) LTraj=min∀k∈K⁡RMSE(Y,Y^),LGoal =RMSE(YregTf,Y^Tf)

where K is the total number of trajectory samples.

For the CVAE loss function LCVAE, the stochastic latent variable Z is sampled from a Gaussian distribution, and the CVAE objective is to maximize its variational lower bound:

(24) maxθ,ϕ,ωEqϕ(Zq|menctc−tobs,E)[logpω(mLatenttc+tpre|menctobs,Ygoaltc+tpre,Z)]−DKL[qϕ(Zq|menctc−tobs,E)∥pθ(Zp|menctc−tobs)]

where the first term maximizes the log-likelihood of the predicted target distribution, and the KL term minimizes the difference between the recognition and conditional prior networks. Finally, the total loss function Ltotal is progressively minimized through backpropagation, adjusting parameters and updating weights iteratively until convergence.

Experimental details

Datasets

To evaluate the effectiveness of the proposed SGANet model, this study utilizes both homogeneous and heterogeneous datasets. JAAD (Rasouli, Kotseruba & Tsotsos, 2017) and PIE (Rasouli et al., 2019) are selected due to their focus on pedestrian behavior in egocentric traffic scenarios, making them ideal for analyzing human-vehicle interactions. Both datasets contain video sequences recorded at 30 FPS from dashboard-mounted cameras, providing high-resolution pedestrian trajectory data. In contrast, the TITAN dataset (Malla, Dariush & Choi, 2020) is chosen for its diverse road user types, including pedestrians, cyclists, motorcyclists, and vehicles, making it suitable for evaluating multimodal trajectory prediction in heterogeneous traffic environments. TITAN is captured using a GoPro Hero 7 with an IMU sensor at 10 Hz (10 FPS), providing synchronized ego-motion data essential for modeling motion dynamics. For JAAD and PIE datasets are split into training, validation, and testing sets, following the 50% overlap approach from Rasouli, Rohani & Luo (2021); For TITAN dataset, we follow the same dataset split as same as TITAN model (Malla, Dariush & Choi, 2020).

Implementation details

The proposed SGANet model was trained on an Ubuntu 20.04 system with an NVIDIA GeForce RTX 3090 GPU for 100 epochs, using the Adam optimizer with a learning rate of 0.0001, a batch size of 128, and a ReduceLROnPlateau scheduler that reduces the learning rate by 0.1 after five epochs without validation loss improvement (minimum threshold: 1e−10). The encoder module processes multimodal inputs, including label class, bounding box coordinates, speed, distance, and IMU/OBD sensor readings, embedding them into 512-dimensional FC layers. The GDF module refines future goal predictions with 128-dimensional goal embeddings, enhanced by MHA with eight attention heads. The CVAE module utilizes a 32-dimensional latent space, with a Gaussian prior (nu = 0.0, sigma = 1.5), generating 20 trajectory samples per forward pass (K = 20). A Bi-LSTM (hidden size = 256) processes future goal locations, while LSTMCells (hidden size = 512) handle sequential dependencies in the trajectory encoder.

To capture socially-aware correlation interaction, SGANet models each road user as a node in a fully connected graph, using GATs to encode interactions. A 1-dimensional social embedding with a single attention head prioritizes neighboring influences, while MHA networks process social interactions. The trajectory encoder integrates bounding box data, goal features, and social interactions, using an LSTMCells (hidden size = 512). The CVAE module consists of a prior network, recognition network, and trajectory generation network. The prior network predicts distribution mean and covariance using goal locations and encoder input features, while the recognition network refines the latent space via a Bi-LSTM (hidden size = 256). The decoder employs MHA (four attention heads) to generate 4D bounding box coordinates for each road user at future timesteps.

Evaluations metrics

The performance of the SGANet model is evaluated using multimodal trajectory prediction, implementing the BoM approaches from K predicted samples. To assess future object localization, we use average displacement error (ADE), final displacement error (FDE), and final intersection over union (FIOU) metrics. ADE and FDE measure the positional error of the bounding box center, while FIOU evaluates the accuracy of the bounding box size and overlap. FIOU is included in the evaluation because ADE and FDE only measure localization error without considering the bounding box dimensions, which are crucial for understanding the spatial accuracy of bounding box predictions. For the TITAN dataset, the model observes 1 s of past data to predict a 2-s future trajectory (Malla, Dariush & Choi, 2020). For the JAAD and PIE datasets, it observes 0.5 s to forecast 1 s ahead (Rasouli, Rohani & Luo, 2021).

Models comparison

We evaluate SGANet on the TITAN dataset (Malla, Dariush & Choi, 2020) alongside several baseline models, such as Social-LSTM (Alahi et al., 2016), Social-GAN (Gupta et al., 2018), Const-Vel (Schöller et al., 2020), TITAN model (Malla, Dariush & Choi, 2020), GPRAR (Huynh & Alaghband, 2021, BiTraP (Yao et al., 2021), ELMA (Li et al., 2022b), CITraNet (Astuti et al., 2024). Then, the proposed SGANet on JAAD (Rasouli, Kotseruba & Tsotsos, 2017) and PIE (Rasouli et al., 2019) alongside with several baseline models that only use unimodal trajectory prediction (K = 1), such as PIE model (Rasouli et al., 2019), BiPed (Rasouli, Rohani & Luo, 2021), PedFormer (Rasouli & Kotseruba, 2023), Attention-LSTM (Lin, Lin & Hung, 2023), CITraNet (Astuti et al., 2024).

Evaluation results

Ablation studies

Table 1 presents the ablation study results on the TITAN dataset (Malla, Dariush & Choi, 2020), evaluating different module integrations using ADE, FDE, and FIOU metrics.

Table 1 Ablation study analysis.

In terms of minADE/minFDE/FIOU. ↓ Donates lower is better. ↑ Donates higher is better.

Exp. n-th	Temporal encoder module	Spatial encoder module	GDF module	CVAE module	minADE↓	minFDE↓	FIOU↑	
1	LSTM	✓	✓		10.66	15.52	0.7151	
2	LSTM		✓	✓	10.97	15.96	0.7123	
3	LSTM	✓		✓	13.97	22.43	0.6235	
4	LSTM	✓	✓	✓	10.25	15.09	0.7369	
5	GRU	✓	✓		12.27	19.20	0.6618	
6	GRU		✓	✓	10.71	15.94	0.7049	
7	GRU	✓		✓	13.83	21.72	0.6131	
8	GRU	✓	✓	✓	10.66	15.86	0.7084	

Ablation studies of temporal module

To ensure fairness, we compare LSTM and GRU networks as temporal encoders using an embedding size of 128 and a hidden size of 256, both integrated with the GDF and CVAE modules. The second and sixth experimental settings reveal that LSTM outperforms GRU with ADE/FDE/FIOU scores of 0.26/0.02/0.0074. This improvement is attributed to LSTM’s ability to store and retrieve diverse information via its cell state and output components, whereas GRU relies on a single hidden state, potentially limiting its capacity.

Ablation studies of spatial encoder module

To evaluate socially-aware interaction modeling, we compare models with and without the GAT network, which assigns influence weights to road users. The second and fourth experiments show that integrating the GAT-based spatial encoder improves ADE/FDE/FIOU by 0.58/0.51/0.0128. This demonstrates that incorporating socially relevant interactions among different road user classes enhances trajectory prediction accuracy.

Ablation studies of GDF module

The third and seventh experiments investigate the impact of the GDF module with both LSTM and GRU encoders. Results indicate that models without GDF perform worse than those incorporating it. The third and fourth comparisons show a significant improvement of 3.58/6.98/0.1016 for ADE/FDE/FIOU when using the GDF module with an LSTM encoder. This highlights how successive goal estimation refines future trajectory prediction, as the GDF module provides continuous and structured goal-driven information.

Ablation studies of CVAE module

The impact of CVAE on multimodal trajectory prediction is assessed by comparing models with and without the CVAE module using LSTM and GRU encoders. Results from the first and fifth experiments reveal that models lacking CVAE underperform, even with different encoders. Specifically, the first and fourth experiments, both employing LSTM encoders, show that incorporating CVAE improves ADE/FDE/FIOU to 0.27/0.07/0.0100. This suggests that CVAE effectively captures trajectory variations, enabling multimodal predictions and enhancing accuracy.

Ablation studies of full SGANet model

The final ablation study evaluates the complete social-goal attention network (SGANet), which integrates spatial-temporal encoding, GDF, and CVAE modules. The fourth and eighth experiments compare SGANet models using LSTM and GRU encoders. Results indicate that the LSTM-based SGANet significantly outperforms its GRU counterpart, achieving ADE/FDE/FIOU of 0.27/0.41/0.0167 improvement.

While simpler models (e.g., without GDF or CVAE) can achieve competitive results, they produce higher prediction errors. For instance, removing GDF worsens ADE by 3.58 and FDE by 6.98, while excluding CVAE increases localization errors due to less effective multimodal capture. Although these simpler architectures reduce computational complexity, they compromise predictive accuracy and robustness. By combining spatial-temporal encoding, GDF, and CVAE, the proposed SGANet strikes the optimal balance, enhancing spatial and temporal interaction modeling, goal-directed forecasting, and multimodal trajectory generation without incurring excessive complexity.

Convergence results

The model’s convergence during training and validation is measured using the total error function defined in Eq. (22). The proposed model was trained for 100 epochs, with loss values stabilizing between 0.3 and 0.4 after approximately 30 epochs, as shown in Fig. 8. This indicates that the model effectively learns trajectory patterns while incorporating multimodal predictions and socially aware interaction correlations, ensuring robust trajectory forecasting across multiple road user classes.

Figure 8 The convergence results of the training and validation process for the proposed SGANet.

The loss stabilizes after approximately 30 epochs, indicating that the model effectively learns trajectory patterns while balancing multimodal prediction, goal-directed, and socially-aware interaction correlations modelling.

Computational cost

Table 2 presents the computational cost of the proposed SGANet, evaluating parameters (PARAMS in M), floating point operations (FLOPs in G), and inference time (seconds). The study compares the computational efficiency of the temporal encoder module with GRU and LSTM networks. While GRU requires fewer gates and parameters, LSTM’s more complex architecture increases inference time. Although SGANet is computationally intensive due to its vision-based multi-stacked neural network design, ablation studies indicate that while GRU offers faster inference, it is less effective and adaptable for trajectory prediction.

Table 2 Computational costs of the proposed method.

The evaluation results donate lower the better.

Method	Temporal encoder networks	PARAMS (M)	FLOPs (G)	Model inference (s)	
Social-goal attention networks (SGANet)	GRU	3.93	171	0.20	
LSTM	4.55	172	0.23	

Socially-aware correlation interaction visualization result

To evaluate our model’s ability to capture implicit socially-aware interaction correlations, we visualize the spatial attention weights learned during training (Fig. 9) using color-coded sequences. Our SGANet employs a GAT to quantify social interactions by dynamically assigning influence weights to different road user classes based on spatial proximity, speed, trajectory direction, and behavioral tendencies. This enables differential weighting of road users, ensuring that the most socially relevant interactions are emphasized in trajectory prediction. For instance, pedestrians at crosswalks or near the ego-vehicle exert a higher influence on trajectory adjustments compared to distant vehicles, while cyclists and motorcyclists require adaptive weighting due to their dynamic maneuverability.

Figure 9 The visualization of socially-aware interaction correlation results.

All figures depict the spatial interaction encoder with varying importance levels of each road user’s influence weight in a scene, experimented on the TITAN dataset. The ten-frame observation sequence is represented with different colors, while the circles illustrate the attention weights, indicating the degree of influence each road user has on the target trajectory prediction.

The visualization highlights how the model prioritizes interactions dynamically, with larger circles representing higher influence levels. The largest circle indicates the road user most critical to the ego-vehicle’s next predicted movement, reflecting the adaptive nature of SGANet’s graph attention mechanism. By assigning higher weights to fast-approaching vehicles and nearby pedestrians while minimizing the impact of stationary objects, SGANet effectively models complex multi-agent interactions. This visualization confirms the model’s ability to intelligently infer and adjust interaction dependencies, leading to more accurate, socially-informed trajectory predictions in dynamic traffic environments.

Quantitative comparison results

Heterogeneous TITAN dataset comparison

Following previous benchmarks, this study uses 10 observed frames to predict 20 future frames, consistent with evaluations of TITAN model (Malla, Dariush & Choi, 2020), ELMA (Li et al., 2022b), and GPRAR (Huynh & Alaghband, 2021) on the TITAN dataset. Table 3 presents a comparative analysis with models such as Social-LSTM (Alahi et al., 2016), Social-GAN (Gupta et al., 2018), and Cons-Vel (Schöller et al., 2020). TITAN-based variants outperform these traditional methods, while GPRAR (Huynh & Alaghband, 2021), which integrates GCN, falls short of TITAN_EP+IP+AP. The ELMA model (Li et al., 2022b) achieves state-of-the-art performance on TITAN with 2.45/2.06 ADE/FDE. Our SGANet, while slightly behind ELMA in ADE (1.18), excels in FDE and FIOU, improving endpoint accuracy by 9.5% and bounding-box accuracy by 6.9%. Meanwhile, CITRaNet model (Astuti et al., 2024) achieves a lower ADE (8.72) than SGANet (10.25), our model significantly outperforms in FDE (15.09 vs. 15.98) and FIOU (0.7369 vs. 0.5919). This due to CITRaNet only focus on person-centric model and its Transformer and Gumbel distribution integration lacks an inherent inductive bias for spatial locality, making them less efficient in encoding fine-grained social interactions compared to GAT-based architectures.

Table 3 Quantitative comparisons on TITAN dataset.

In terms of minADE/minFDE/FIOU. ↓ donates lower is better. ↑ donates higher is better.

Methods (Best of 20)	Year	minADE↓	minFDE↓	FIOU↑	
Social-LSTM	2016	37.01	66.78	–	
Social-GAN	2018	35.41	69.41	–	
Const-Vel	2020	44.39	102.47	0.1692	
Titan_vanilla	2020	38.56	72.42	0.3233	
Titan_AP	2020	33.54	55.80	0.3670	
Titan_EP+IP+AP	2020	11.32	19.53	0.6559	
GPRAR	2021	12.56	20.36	–	
ABC+	2022	30.52	46.84	–	
ELMA	2022	9.21	16.53	–	
CITRaNet	2024	8.72	15.98	0.5919	
SGANet	–	10.25	15.09	0.7369	

To evaluate SGANet’s effectiveness across both short and long prediction horizons, Table 4 provides additional results. For short-term predictions (2 s, 20 frames at 10 FPS), we maintain the setup used in Table 3. To assess long-term performance, we extend predictions to 30, 40, 50, 60, and 70 frames. Results indicate that error rates increase with longer prediction horizons, primarily due to the accumulation of uncertainties over time. This highlights the challenge of long-term trajectory forecasting and the importance of capturing socially-aware interactions for robust predictions.

Table 4 The comparison results in various durations over short to long terms trajectory predictions using minADE ↓/minFDE ↓ metrics.

Predicted durations	Social-goal attention networks models (SGANet)	
2 s	10.25/15.09	
3 s	12.77/21.97	
4 s	14.56/25.75	
5 s	19.04/40.06	
6 s	23.74/45.87	
7 s	25.67/49.72	

Homogeneous JAAD and PIE datasets comparison

Table 5 presents a comparative analysis of trajectory prediction performance across multiple models using ADE and FDE. Unlike previous unimodal and deterministic models, SGANet incorporates multimodal trajectory prediction, allowing it to capture uncertainties, trajectory variations, and socially-aware interactions more effectively. The PIE Model (Rasouli et al., 2019) exhibits the highest errors (ADE: 22.83, FDE: 49.44), as it relies on deterministic trajectory forecasting without accounting for multimodal variations. BiPed (Rasouli, Rohani & Luo, 2021), which introduces multitask learning for joint trajectory and action prediction, improves over PIE but still struggles with capturing multimodal uncertainty, leading to ADE and FDE errors of 20.58 and 46.85, respectively. SGANet further reduces these errors by 63.2% in ADE and 66.3% in FDE, confirming the advantage of multimodal trajectory prediction over deterministic multitask approaches.

Table 5 Quantitative comparisons on JAAD and PIE dataset.

In terms of ADE/FDE/FIOU. ↓ donates lower is better.

Methods	Year	ADE↓	FDE↓	ADE↓	FDE↓	
PIE model	2019	22.83	49.44	19.50	45.27	
BiPed	2021	20.58	46.85	15.21	35.03	
PedFormer	2023	17.89	41.63	13.08	30.35	
Attention-LSTM	2023	–	–	9.97	19.43	
CITRaNet	2024	16.64	37.65	8.50	18.21	
SGANet (K = 20)	–	9.49	15.76	6.92	10.92	

PedFormer (Rasouli & Kotseruba, 2023) employs a cross-modal Transformer network, enhancing pedestrian-vehicle interaction modeling but limited to spatial-temporal inductive biases. While it reduces errors compared to BiPed (ADE: 17.89, FDE: 41.63), SGANet still achieves a 73.3% and 74.1% error reduction, respectively, highlighting the strength of GAT-based socially-aware interaction modeling over Transformer-based approaches. Attention-LSTM (Lin, Lin & Hung, 2023) integrates attention mechanisms with LSTM, improving short-term trajectory prediction (FDE: 19.43) but SGANet still can reduce ADE and FDE errors by 30.6% and 43.8%, respectively, demonstrating its effectiveness in capturing goal-directed modelling. CITRaNet (Astuti et al., 2024), which employs a Transformer-based Gumbel distribution model, achieves a competitive ADE of 16.64 but underperforms in FDE (37.65), suggesting instability in endpoint prediction. SGANet significantly reduces ADE error by 57% and FDE error by 58.2%, indicating that GAT-based socially-aware multimodal predictions are more effective than Transformer-Gumbel architectures for pedestrian trajectory forecasting.

Qualitative results

Heterogeneous TITAN dataset visualizations

Figure 10 illustrates the observed (cyan), ground truth (red), and predicted (green) trajectories in various driving scenarios. This study presents four qualitative trajectory predictions: (a) and (b) depict multi-target road user predictions, while (c) and (d) focus on a single target’s predicted destination and bounding box. The multimodal trajectory distribution is visualized through 20 sampled predictions (e, f) and KDE-based density estimations (g, h). These results indicate that SGANet enhances prediction robustness, helping autonomous vehicles anticipate sudden changes and accurately estimate road users’ movement directions.

Figure 10 The qualitative results of four different visualization types.

(A–D) illustrate unimodal visualizations, where a single predicted trajectory is shown for each road user, while (E–H) illustrate multimodal visualizations, where multiple possible trajectories are generated to capture trajectory variations. The visualization highlights how the model accounts for trajectory uncertainty and socially-aware interactions. Note in colors: green, red, and blue respectively represent the predicted, ground truth, and observed trajectories.

Figure 11 further examines trajectory predictions over extended time horizons (2 to 7 s). As shown, longer prediction intervals (represented by dot markers) result in greater deviation from the ground truth, especially beyond 6 s. In first-person view (FPV) datasets, extended predictions lead to smaller road users being filtered out, as seen in Fig. 11, where only one road user remains visible after 70 frames. These findings highlight the challenge of long-term trajectory forecasting due to accumulated uncertainty and occlusion effects.

Figure 11 The qualitative results of different predicted frames ranging from 2 to 7 s at 10 FPS, with dot markers indicating each time step.

The visualization highlights how prediction accuracy varies across different time horizons, where shorter predictions align more closely with the ground truth, while longer predictions exhibit increased deviation due to accumulated uncertainty.

Homogeneous JAAD and PIE datasets visualization

Figure 12 illustrates the visualization results on the JAAD (A), (B), (C) and PIE (D), (E), (F) datasets, highlighting the multi-target pedestrian trajectory predictions generated by the proposed SGANet. The visualization demonstrates SGANet’s ability to model pedestrian movement across diverse urban scenarios, accurately capturing spatial interactions, pedestrian behaviors, and trajectory uncertainties. By leveraging socially-aware correlations, goal-directed forecaster, and multimodal trajectory prediction, the model can anticipate the robustness in predicting pedestrian trajectories under varied lighting conditions, occlusions, and urban complexities. Notably, the predicted trajectories align closely with the ground truth, validating the model’s capacity to generate reliable, socially compliant, and context-aware predictions. These results suggest that SGANet is effective in improving safety-critical decision-making for autonomous vehicles by reducing uncertainty in pedestrian intent estimation and future trajectory forecasting.

Figure 12 The qualitative results of multi-target pedestrian trajectory predictions on JAAD (A–C) and PIE (D–F) datasets.

Note in colors: green, red, and blue represent the predicted, ground truth, and observed trajectories, respectively. The visualization demonstrates the model’s capability to accurately forecast pedestrian movements in various traffic scenarios, capturing their future paths with consideration of socially-aware interactions.

Extreme and complex scenario visualization

Figure 13 presents a sequence of successive frames showcasing multimodal trajectory predictions using Gaussian KDE in diverse and dense urban scenarios. The visualization highlights the model’s ability to generalize across varying traffic conditions, including complex interactions between pedestrians, cyclists, and vehicles. The predicted trajectories capture multiple possible trajectory patterns, effectively handling uncertainties in unpredictable pedestrian behavior and high-density urban traffic. In complex scenarios, such as crowded intersections or sudden pedestrian crossings, the model maintains robust performance by leveraging socially-aware interaction correlations and goal-directed forecasting. Additionally, the CVAE module enables the model to capture variations in driving styles and pedestrian movement unpredictability by generating diverse trajectory samples based on individual behavioral tendencies. This ensures that different reaction times, acceleration patterns, or abrupt pedestrian crossings are better accounted for in the trajectory prediction process. These qualitative results complement the reported ADE, FDE, and FIOU metrics, reinforcing the model’s adaptability and accuracy in extreme traffic situations. These dynamic visualizations further demonstrate the model’s ability to maintain accurate trajectory predictions over successive frames, adapting to diverse traffic conditions and complex interactions in real-world urban scenarios.

Figure 13 The qualitative results of multimodal trajectory predictions using Gaussian KDE in successive frames.

The figure illustrates predicted trajectories in complex urban scenarios, where multiple road users interact dynamically. Gaussian kernel density estimation (KDE) is applied to visualize multimodal trajectory distributions, capturing uncertainties in diverse motion patterns.

Conclusions and future work

This article presents social-goal attention networks (SGANet), a vision-based framework for forecasting the trajectories of multiple road users by incorporating socially-aware interactions, goal-directed forecaster, and multimodal predictions modelling. The model integrates a spatial-temporal encoder, where graph attention networks (GATs) capture spatial interactions and long short-term memory (LSTM) networks model temporal dependencies. A goal-directed forecaster (GDF) predicts the end point destinations, while a conditional variational autoencoder (CVAE) generates multimodal trajectory distributions. The decoder, equipped with multi-head attention (MHA) and feed-forward networks (FFN), interprets CVAE outputs for trajectory refinement. Evaluated on the both homogeneous (JAAD and PIE) and heterogeneous TITAN datasets, SGANet outperforms previous benchmarks in trajectory prediction accuracy.

One limitation of SGANet is that GAT, designed to model social interactions per frame, supports only up to 50 road users, restricting its scalability in highly crowded environments. To address this, we plan to develop a dynamically adjustable storage matrix that enables tracking more road users, enhancing adaptability in complex traffic scenarios. Additionally, we intend to curate a dataset of hazardous traffic situations in Taiwan, which presents greater challenges than existing public datasets, to further evaluate and refine SGANet’s performance.

Additional Information and Declarations

Competing Interests

The authors declare that they have no competing interests.

Author Contributions

Lia Astuti conceived and designed the experiments, performed the experiments, analyzed the data, performed the computation work, prepared figures and/or tables, authored or reviewed drafts of the article, and approved the final draft.

Chui-Hong Chiu performed the experiments, analyzed the data, performed the computation work, prepared figures and/or tables, authored or reviewed drafts of the article, and approved the final draft.

Yu-Chen Lin conceived and designed the experiments, authored or reviewed drafts of the article, and approved the final draft.

Ming-Chih Lin conceived and designed the experiments, performed the experiments, analyzed the data, performed the computation work, prepared figures and/or tables, and approved the final draft.

Data Availability

The following information was supplied regarding data availability:

The data is available at the TITAN from Honda Research Institute USA: https://usa.honda-ri.com/titan.

Our code is available at Zenodo: Yu-Chen Lin. (2025). Socially-Aware-Trajectory-Prediction. Zenodo. https://doi.org/10.5281/zenodo.14957604.

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
