# Peer review of "Social-aware trajectory prediction using goal-directed attention networks with egocentric vision"

_PeerJ Computer Science, doi:10.7717/peerj-cs.2842_

## Round 0.1 · original submission · Major Revisions

Thank you for your submission to PeerJ. After a thorough peer-review process, the reviewers have recognized the valuable contributions of your work, particularly the integration of multiple neural network modules for road agent trajectory prediction using egocentric vision. However, they have identified several areas that require substantial improvements before the manuscript can be considered for publication.

Key Areas for Improvement

1. Clarity of Motivation and Justification of the Proposed Approach
o A more detailed explanation of the research gaps your study addresses and how your method specifically tackles them is needed.
o Strengthen the justification for selecting the TITAN dataset and consider additional experiments on widely used datasets such as PIE and JAAD.
2. Enhancement of Methodological Details
o Provide a more detailed description of the model training process, including the selection of hyperparameters and optimization strategies.
o Elaborate on how different data streams (e.g., dashboard camera and IMU sensor) are synchronized and processed within the proposed model.
o Clarify how social interactions between different road users are prioritized or weighted in the prediction process.
3. Experimental Evaluation and Performance Analysis
o Expand the discussion on the model’s performance in various scenarios, including high-density urban traffic or unpredictable pedestrian behavior.
o Justify the trade-off between model complexity and predictive accuracy by discussing whether simpler models could achieve similar results.
o Provide additional qualitative scenario-based evaluations to support the reported quantitative metrics (ADE, FDE, FIOU), especially in complex or extreme traffic situations.
4. Language and Presentation Improvements
o The clarity of expression needs improvement to ensure that an international audience can fully understand the manuscript. The reviewers have pointed out multiple grammatical errors and ambiguous phrasing that should be revised.
o Figures should be replaced with vector-based versions to improve clarity and readability.
o Ensure that figure and table descriptions are precise and clearly reference their corresponding parts.

Decision: Major Revisions

Given the extent of the required revisions, we are inviting you to submit a revised version of the manuscript addressing all reviewer comments. Please provide a detailed response letter explaining how each comment has been addressed. If certain suggestions could not be incorporated, justify why.

We appreciate your efforts and look forward to your revised submission.

Reviewer 1 ·

Basic reporting

This paper proposed a method for trajectory prediction. The model integrates social interaction, goal features and so on to produce future trajectory of agents. The expression is clear. The paper conforms to PeerJ standards. However, it has the following problems.
(1) The motivation is not very clear. Please explain the current research gaps in detail.
(2) Please explain how the proposed method deal with the above gaps.
(3) Many figures in the paper are not clear. They should be replaced with vector figures.

Experimental design

In terms of experiments, I have the following comments.
(1) Why the authors select TITAN dataset to verify their approach.
(2) How is the performance of their approach on more popular datasets such as PIE and JAAD.
(3) Please add more recent approaches in experiments for comparison.

Validity of the findings

The paper can be viewed as an incremental work in trajectory prediction.

Reviewer 2 ·

Basic reporting

This paper proposes social-goal attention networks with vision-based multi-stacked neural
network integrations for multiple classes of road user trajectory prediction task.It has innovation and practical value.However, the major problem with the article is that the expression is not clear enough, and there are multiple grammar errors in the English language that need to be carefully revised.

Experimental design

The authors conduct extensive experiments on the TITAN dataset and compare it with existing benchmark models.The experimental results show that the proposed model outperforms the baseline model in certain indicators.

Validity of the findings

The authors provide a detailed description of the architecture of the model and the functions of each component, including spatial interaction encoder, temporal encoder, GDF module, and CVAE module. These descriptions help to understand the working principle of the model.

Additional comments

The authors should consider the following concerns:
1. The authors provide a detailed description of the architecture of the model and the functions of each component, including spatial interaction encoder, temporal encoder, GDF module, and CVAE module. These descriptions help to understand the working principle of the model. However, I suggest the authors provide more details about the model training process, such as the selection of hyperparameters and optimization strategies.
2. The authors conduct extensive experiments on the TITAN dataset and compare it with existing benchmark models.The experimental results show that the proposed model outperforms the baseline model in certain indicators. But I suggest the authors further discuss the performance of the model in different scenarios, as well as the robustness and generalization ability of the model.
3. Please carefully check the description of the figure and table and use clear language as much as possible.For example, in the description of Figure 10, “The first and second rows illustrate unimodal visualizations and the third and fourth rows” does not clearly indicate corresponding part in the figure.
4. The English language should be improved to ensure that aninternational audience can clearly understand your text. Some examples where the language could be improved include lines 246, 268, 269, 282, 283, 284, 300, 368 – the current phrasing makes comprehension difficult. I suggest you have a colleague who is proficient in English and familiar with the subject matter review your manuscript, or contact a professional editing service.

Reviewer 3 ·

Basic reporting

This study presents a novel social-goal attention network designed for trajectory prediction of road agents using egocentric vision. The model integrates multiple neural network modules to account for social interactions, temporal dependencies, and goal-directed behaviors in predicting uncertain future trajectories. My comments are as follows.
1. The data are collected by the dashboard camera and IMU sensor. How are these two data streams synchronized and processed within the proposed model? A more detailed explanation would be helpful for understanding the robustness of the model.
2. How well does the model generalize to diverse traffic conditions, such as high-density urban traffic or complex, unpredictable pedestrian behaviors? Please add some explanations.
3. The proposed model is relatively complex. While the results on the TITAN dataset are promising, the paper could further explore the trade-off between the model's complexity and its performance. Specifically, does the increased model complexity justify the improvements in prediction accuracy, or are there simpler models that could achieve similar results?

Experimental design

4. The process of concatenating the different input sequences (bounding box coordinates, speed, odometry data, etc.) and how these are aligned temporally with the observed history (up to the current frame) is not entirely clear. Please make this part more explicit
5. A concept of "socially-aware correlation interaction model" is introduced in this paper, which is innovative. But I was still wondering how these socially-aware features, particularly interactions between multiple road users are modeled in the trajectory prediction. For example, how does this model prioritize or weigh interactions between certain classes of road users differently in the prediction process?

Validity of the findings

6. When evaluating ADE/FDE/FIOU, the paper only mentions some quantitative metrics, but lacks specific scenario examples to support these evaluations. It is recommended to include an analysis of several typical scenarios in the results section to illustrate the performance differences of these metrics in different traffic scenarios, helping readers to more intuitively understand the model's actual performance. For example, the model's prediction performance could be demonstrated in complex or extreme scenarios, such as dense traffic or emergency situations.

Additional comments

7. Although GNN can effectively capture temporal and spatial dependencies, it may not fully account for individual differences between drivers, such as driving styles, reaction times, etc. CVAE, through the conditional generative model, can generate trajectories that align with the specific characteristics of a driver based on their personal traits. Therefore, the authors should discuss articles related to driving styles in the related works section.

---

## Round 0.2 · accepted · Accept

All the comments suggested by reviewers have been addressed.
Congratulations!

Reviewer 1 ·

Basic reporting

The authors have addressed all my concerns.

Experimental design

The experiments are sufficient.

Validity of the findings

The proposed method is interesting.

Reviewer 2 ·

Basic reporting

The paper has been revised well.

Experimental design

The paper has been revised well.

Validity of the findings

The paper has been revised well.